# NAVIGATE: Evaluating Visual-Guided Search Decision-Making on the Open Web

**YaoQi Fan** [1][*]  **Zhe Chen** [1][*]  **Wei Zhu** [2]  **Kangxin Yin** [1]  **Yangzhou Liu** [1]  **Yue Cao** [1]  **Zhi Zhu** [1]  **Tong Lu** [1]

## Abstract

Vision–Language Models (VLMs) are increasingly deployed with web search tools, yet we still lack benchmarks that isolate a critical capability for real-world use: deciding when to search and how to steer search from ambiguous visual evidence, especially when multiple images provide overlapping or conflicting cues. We present **NAVIGATE**, a novel benchmark centered on images as primary evidence for open-web search planning and multi-step reasoning. It contains 500 questions across 20 domains and spans three difficulty tiers, from single-image, self-contained problems to multi-image joint search and multi-domain composition. Unlike prior benchmarks that specify explicit search targets, NAVIGATE evaluates search decision-making: models must infer whether external search is necessary and iteratively refine search directions based on holistic reasoning over visual cues. Across a broad set of VLMs and search-enabled systems, performance remains low, Gemini-3-Pro-Preview-Search reaches only 36.4% accuracy, highlighting persistent failures in cross-image grounding, search triggering, and search strategy coordination. We release the dataset at https://github.com/fantupang/NAVIGATE.

## 1. Introduction

Many recent vision–language model (VLM) systems (Bai et al., 2025b; Chen et al., 2024b; Bai et al., 2025a; OpenAI, 2025a; ByteDance Seed, 2025; Comanici et al., 2025) integrate web search tools, enabling them to formulate search queries, retrieve image and text evidence, and synthesize

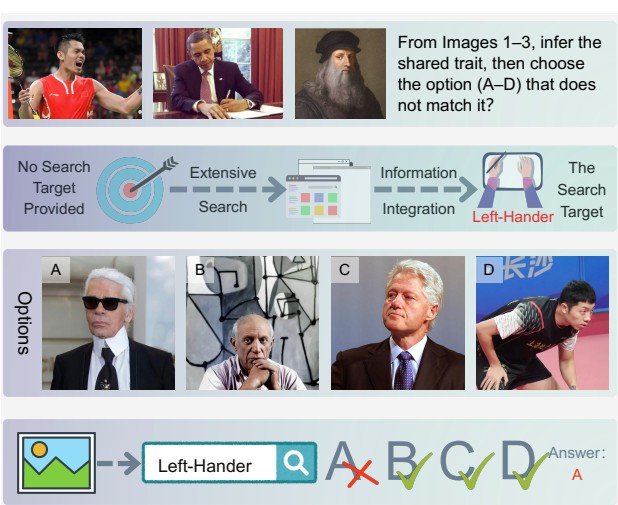

*Figure 1.* NAVIGATE targets visual-guided search decision-making beyond search execution. Given multiple images and no explicit search target, models must infer an implicit goal from distributed visual cues, decide whether web search is warranted, and steer search to gather and integrate open-web evidence. The "left-hander" example is one case; NAVIGATE covers diverse image-centric scenarios across domains and difficulty levels.

answers from online content. However, real failures often stem from search decision-making rather than search execution, specifically deciding when to search and how to steer it. This challenge is particularly acute for image-centric scenarios, where the target is implicit and must be inferred from incomplete, or even conflicting cues across images. Such inputs are common in everyday workflows, *e.g.*, pairing a product page with a label photo, sharing screenshots that capture different steps of an issue, or providing a document snippet with supporting evidence. As a result, models may over-search and drift to irrelevant results, under-search and miss key evidence, or commit to a plausible but incorrect trajectory by anchoring on a local cue.

Existing search and browsing benchmarks (Phan et al., 2025; Wei et al., 2025; Zhou et al., 2025; Li et al., 2025; Jiang et al., 2024) are challenging, but they do not fully capture decision-making in visually grounded, multi-image settings. In many cases, text provides the main search cues and images, when present, are auxiliary. What remains missing is

[1]State Key Laboratory for Novel Software Technology, Nanjing University, Nanjing, China [2]China Mobile Zijin Innovation Institute, Nanjing, China. Correspondence to: Tong Lu <lutong@nju.edu.cn>.

*Proceedings of the 43rd International Conference on Machine Learning*, Seoul, South Korea. PMLR 306, 2026. Copyright 2026 by the author(s).

*Table 1.* Comparison of existing search benchmarks (Mialon et al., 2023; Li et al., 2025; Wei et al., 2025; Jiang et al., 2024; Phan et al., 2025; Tao et al., 2025; Fu et al., 2025) by modality, search requirements, and dataset statistics. "Elective" denotes whether the model must decide whether to invoke search (*i.e.*, the when-to-search decision). Attributes follow the official papers; unspecified cases are marked "–".

| Benchmark | Modality | | Search Characteristics | | | | Dataset Statistics | | |
|---|---|---|---|---|---|---|---|---|---|
| | Images | Multi-Image | Multi-Hop | Cross-Domain | Implicit Target | Elective | #Sample | #Img$_{avg}$ | #Domain |
| MMSearch-Plus | ✓ | ✓ | ✓ | ✗ | – | ✗ | 311 | – | 43 |
| GAIA | ✓ | ✗ | ✓ | ✗ | ✗ | ✗ | 466 | – | – |
| BrowseComp | ✗ | ✗ | ✓ | ✗ | ✗ | ✗ | 1266 | 0 | 10 |
| MM-BrowseComp | ✓ | ✗ | ✓ | ✗ | ✗ | ✗ | 224 | 1.02 | 22 |
| HLE-VL | ✓ | ✗ | ✗ | ✗ | ✗ | ✗ | 350 | 1 | 8 |
| MMSearch | ✓ | ✗ | ✗ | ✗ | ✗ | ✗ | 300 | 0.54 | 14 |
| LiveVQA | ✓ | ✗ | ✓ | ✗ | ✗ | ✗ | 1232 | 1 | 14 |
| NAVIGATE (ours) | ✓ | ✓ | ✓ | ✓ | ✓ | ✓ | 500 | 4.62 | 20 |

a systematic evaluation of whether a model can (1) decide whether external search is warranted from visual cues, (2) revise search directions as cross-image evidence accumulates, and (3) integrate evidence across images and domains into a coherent reasoning trajectory. We further compare our benchmark with existing search-oriented benchmarks, as summarized in Table 1.

To bridge this gap, we propose NAVIGATE, a benchmark for evaluating visual-guided search decision-making on the open web. NAVIGATE centers images as primary cues and measures whether models can decide when to invoke external search and how to steer it through multi-step searching and reasoning (see Figure 1). Unlike existing benchmarks (Zhang et al., 2024; Yue et al., 2024; Wang et al.; Fu et al., 2024; Cheng et al., 2025), which are often solvable using parametric knowledge alone, NAVIGATE requires iterative evidence gathering from open-web sources grounded in visual inputs. The benchmark spans realistic settings from single-image questions that penalize unnecessary search, to multi-image and multi-domain problems that require cross-image cue integration and systematic search to uncover latent relationships across disparate contexts. To keep evaluation transparent and reproducible, we formulate search decision-making as an end-to-end QA task, with the core signal encoded in carefully designed instances.

To assess the difficulty and diagnostic value of NAVIGATE, we evaluate a diverse set of VLMs (OpenAI, 2025a;b; Comanici et al., 2025; Bai et al., 2025a; Chen et al., 2024b; Wang et al., 2025) and search-enabled systems (Google DeepMind, 2025; Comanici et al., 2025; OpenAI, 2025b; xAI, 2025; Team et al., 2025b; ByteDance Seed, 2025) on NAVIGATE. Results show that NAVIGATE is highly challenging and diagnostic: even strong search-augmented models frequently fail to decide when search is warranted (including search-avoidance cases) and to steer multi-step search from ambiguous multi-image cues, with performance dropping further in multi-domain compositions, highlight-

ing substantial room for improvement in visual-guided search decision-making on the open web.

Our main contributions are summarized as follows:

- We introduce NAVIGATE, a benchmark for visual-guided search decision-making on the open web. It stresses image-centric ambiguity, requiring models to decide whether to invoke search and how to steer multi-step search from distributed, overlapping, or conflicting visual cues, including search-avoidance cases where browsing is unnecessary.

- We propose a failure-driven collection paradigm grounded in realistic search interactions, where initial answers fail and are corrected through query refinement or intervention. We further curate the dataset to retain instances that are informative for evaluation.

- We evaluate a wide range of VLMs and search-enabled systems and introduce structure-aware scoring for multi-choice and ranking questions (*e.g.*, coverage and ordering consistency) to move beyond binary accuracy and enable fine-grained analysis of search behaviors.

## 2. NAVIGATE

### 2.1. Overview

NAVIGATE is an image-centric benchmark for evaluating visual-guided open-web search decision-making. It targets realistic scenarios where the information is implicit and must be inferred from visual evidence, often distributed across multiple images and occasionally noisy or conflicting. Models must decide whether external search is necessary, and when it is, how to steer search via iterative query refinement while staying grounded in the visual cues.

Motivated by these, NAVIGATE contains 500 curated questions spanning 20 domains, 4 task types, and 3 difficulty

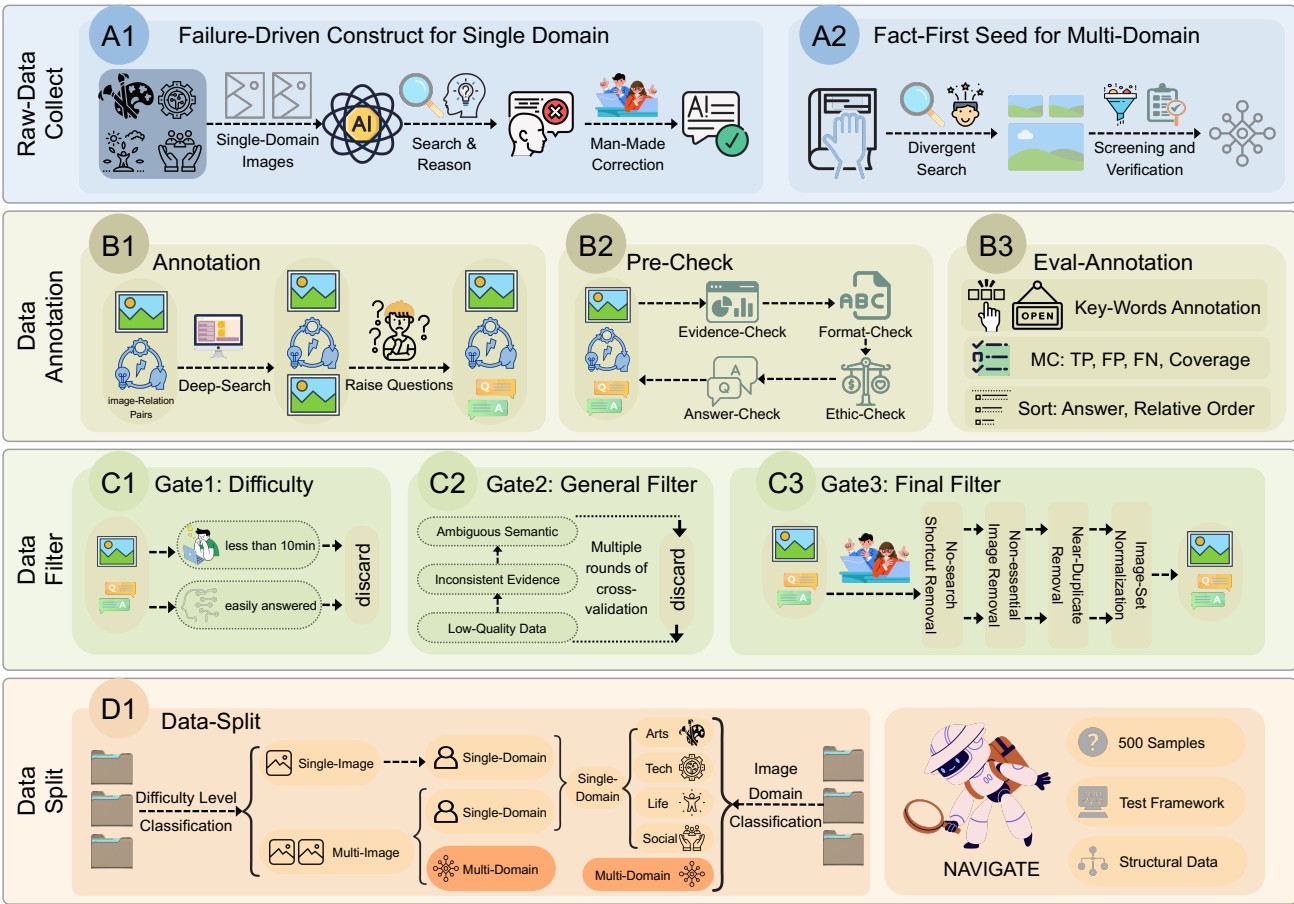

*Figure 2.* The dataset construction pipeline of NAVIGATE. We build the data through two complementary sources: (A1) a failure-driven collection to collect single-domain data, and (A2) a fact-first collection to collect multi-domain data. The candidates are then annotated by forming image-relation pairs and writing questions/answers (B1), and pre-checked for evidence consistency, answer correctness, format validity, and ethical issues (B2). Finally, we apply a three-stage filter—difficulty gate, general quality gate, and final filter, splitting the remaining samples by difficulty, domain, and single-/multi-domain settings to produce the final benchmark.

tiers, ranging from visually self-contained single-image questions to multi-image joint search and multi-domain composition. The benchmark is built with a failure-driven construction and filtering pipeline that preserves real search dynamics such as search failures, path corrections, and strategy variations, while keeping each instance concise and automatically evaluable. We release the dataset at `https://github.com/fantupang/NAVIGATE`

## 2.2. Design Principles

**Evidence-Grounded Search Constraint.** As a benchmark for open-web search decision-making, NAVIGATE enforces an evidence-grounded constraint: every question must be answerable only by grounding in the provided images and, when needed, verifiable evidence searched from the web. We keep textual prompts intentionally lightweight so that models cannot rely on language priors or parametric recall without visual grounding. Crucially, NAVIGATE

includes both search-required instances, where decisive evidence is not contained in the images and must be obtained through external search, and search-avoidance instances, typically single-image questions, where the images alone are sufficient and unnecessary browsing is penalized. To further reduce "lucky" answers from memorization, we prioritize long-tail facts and compositional relations that are unlikely to appear verbatim in training data.

**Answer Uniqueness and Verifiability.** To ensure stable evaluation and reliable scoring, NAVIGATE adopts objective question formats, including single-choice, multi-choice, sorting, and short open-ended questions. Answers are constrained to fewer than four words, reducing ambiguity and improving grading consistency. During annotation, we adopt a fact-first strategy for multi-domain questions (see Figure 2 (A2)): we start from a predefined fact with a unique answer and collect supporting evidence from multiple web sources and multiple images. While multi-image cues may

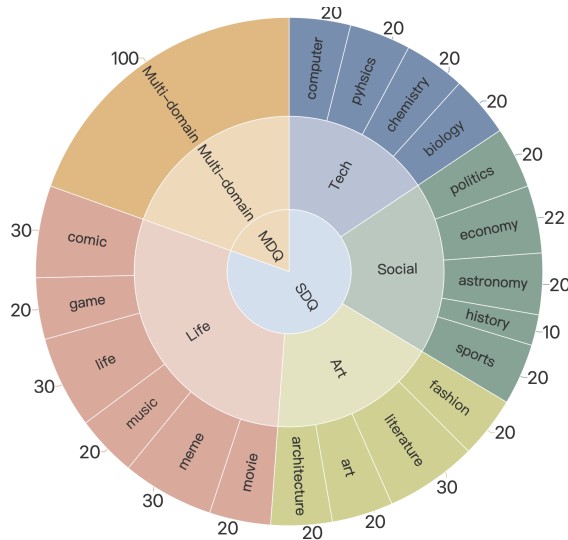

*Figure 3.* Data distribution of NAVIGATE, which contains 500 questions spanning 20 domains, 4 task types, and 3 difficulty tiers.

suggest multiple plausible search directions, we only retain instances whose evidence *converges* to a single independently verifiable answer; annotators additionally attempt to surface alternative verifiable answers or contradictory evidence, and we discard any instance that admits multiple defensible final answers. For time-sensitive queries, we record timestamps and archive the supporting evidence (e.g., a page snapshot) to fix the referenced fact and avoid ambiguity from later updates.

### 2.3. Data Construction and Filtering

**Failure-Driven Data Construction.** As shown in Figure 2(A1), NAVIGATE adopts a failure-driven data construction paradigm rather than a knowledge-centric one. Except for multi-domain questions built via divergent search from a single seed fact, the remaining instances are derived from error-correction trajectories observed in real-world VLM usage or high-fidelity simulations. We collect search interaction logs and select cases where initial predictions are incorrect yet can be corrected through human guidance and refined search trajectories, aligning with NAVIGATE's goal of diagnosing search failures (details provided in Appendix A.3). As a result, NAVIGATE reflects a structured projection of real-world failure-and-correction processes rather than a static knowledge-based QA distribution.

**Data Filtering Strategy.** We first remove ambiguous, ill-defined, or overly trivial questions (Figure 2(C1–C2)), reducing the candidate pool from 1,800 to 1,031. Next, we apply a series of model-agnostic filtering criteria to ensure that each remaining instance admits a clear and verifiable solution path. We discard questions that do not require genuine

visual grounding. For search-required questions, we additionally remove those solvable without external search; for single-image questions, whose information is self-contained, we verify that the image evidence alone suffices. We further eliminate redundant images and retain only the minimal visual set needed for the decision. We further filter out near-duplicate questions and normalize image sets to retain only the minimal set of visual inputs necessary for solving the task, thereby avoiding spurious cues and unintended shortcuts (see Figure 2(C3)). Finally, from the filtered pool, we curate a balanced subset of 500 instances across domains and difficulty levels, ensuring broad coverage while maintaining consistent annotation quality.

### 2.4. Data Split

**Task Categorization by Data Source.** Based on the knowledge sources required by each question and their associated domains, NAVIGATE categorizes tasks into four major groups and twenty domains (see Figure 3). This categorization covers a wide range of real-world search scenarios while enabling fine-grained characterization of model performance across different problem types, thereby facilitating systematic analysis of model strengths and weaknesses in specific domains and task settings.

**Search-Necessity-Aware Difficulty Grading.** Rather than treating difficulty solely as a function of the number of images or the breadth of knowledge involved, NAVIGATE further incorporates search necessity as an explicit factor in difficulty grading. Based on both visual compositionality and the role of external search, we categorize questions into three levels: visually self-contained single-image questions, multi-image single-domain questions, and multi-image multi-domain questions (see Figure 2(D1)).

Single-image questions are visually self-contained—the answer can be derived from the image without consulting external sources. For these instances, invoking web search is not only unnecessary but can also introduce irrelevant or conflicting evidence, which often degrades final performance. We treat them as a diagnostic control subset, explicitly assessing whether a model can recognize when search is unwarranted and refrain from calling search tools.

Multi-image single-domain questions require models to integrate evidence across multiple images within the same knowledge domain and represent the largest portion of the dataset. These questions assess models' ability to coordinate search depth with cross-image evidence aggregation. Multi-image multi-domain questions involve cross-domain reasoning, placing the highest demands on models in terms of search breadth, contextual grounding, and reasoning.

*Table 2.* Accuracy breakdown by task type, difficulty level, and image domain. Models are grouped by whether they support search capability. Task types: MCQ = multi-choice question, SCQ = single-choice question, OEQ = open-ended question, Sort = sorting. Difficulty levels: SIQ = single-image question, MIQ = multi-image question, MDQ denotes both domain and difficulty classification.

| Model | Overall | Task-Type | | | | Difficulty-Level | | | Image-Domain | | | | |
|---|---|---|---|---|---|---|---|---|---|---|---|---|---|
| | | MCQ | OEQ | SCQ | Sort | SIQ | MIQ | MDQ | Arts | Tech | Life | Social | MDQ |
| **VLMs with Search** | | | | | | | | | | | | | |
| Gemini-3-Pro-Preview-Search | 36.40% | 41.60% | 21.23% | 55.78% | 44.90% | 55.56% | 33.58% | 29.23% | 29.55% | 46.91% | 42.47% | 35.44% | 29.23% |
| Gemini-2.5-Flash-Search | 25.10% | 28.00% | 10.06% | 42.18% | 38.78% | 33.33% | 22.41% | 19.40% | 14.77% | 25.93% | 34.25% | 32.91% | 19.40% |
| GPT-5.1-Search | 34.11% | 32.23% | 23.91% | 54.05% | 39.58% | 26.32% | 31.25% | 25.37% | 39.33% | 37.21% | 33.57% | 31.51% | 25.37% |
| GPT-5-Search | 33.26% | 36.36% | 16.85% | 52.70% | 41.67% | 36.84% | 30.22% | 23.88% | 33.71% | 38.37% | 31.43% | 35.62% | 23.88% |
| Grok-3-DeepSearch | 22.18% | 16.80% | 12.29% | 41.50% | 24.49% | 22.22% | 18.86% | 16.92% | 21.59% | 25.93% | 26.03% | 20.25% | 16.92% |
| GLM-4.5V | 15.16% | 14.88% | 6.52% | 27.03% | 22.92% | 47.37% | 10.42% | 4.48% | 13.48% | 23.26% | 12.14% | 21.92% | 4.48% |
| Seed-1.8-251228 | 21.68% | 22.31% | 10.87% | 38.51% | 29.17% | 31.58% | 17.62% | 14.93% | 29.21% | 26.74% | 15.00% | 24.66% | 14.93% |
| Claude-Sonnet-4.5-250929 | 18.74% | 19.01% | 13.04% | 31.08% | 27.08% | 31.58% | 14.28% | 13.43% | 17.98% | 26.74% | 14.29% | 23.29% | 13.43% |
| Perplexity-Sonar | 9.47% | 4.96% | 6.52% | 19.59% | 4.17% | 21.05% | 4.77% | 4.48% | 8.99% | 11.63% | 10.00% | 9.59% | 4.48% |
| **VLMs without Search** | | | | | | | | | | | | | |
| GPT-5.1 | 28.42% | 32.23% | 17.93% | 42.57% | 37.50% | 42.11% | 22.76% | 17.91% | 29.21% | 36.05% | 24.29% | 23.29% | 17.91% |
| GPT-5 | 27.16% | 27.27% | 16.30% | 45.95% | 33.33% | 42.11% | 22.76% | 16.42% | 32.58% | 37.21% | 18.57% | 27.40% | 16.42% |
| Gemini-2.5-Pro | 24.27% | 28.00% | 13.97% | 39.46% | 34.69% | 27.78% | 21.90% | 7.69% | 22.73% | 23.46% | 30.14% | 29.11% | 7.69% |
| Qwen3-VL-235B-A22B-Thinking | 17.89% | 20.66% | 11.41% | 28.38% | 25.00% | 42.11% | 13.25% | 8.96% | 20.22% | 23.26% | 12.14% | 24.66% | 8.96% |
| Qwen3-VL-235B-A22B-Instruct | 16.84% | 17.36% | 8.70% | 29.05% | 22.92% | 47.37% | 12.48% | 4.48% | 20.22% | 20.93% | 14.29% | 21.92% | 4.48% |
| OpenAI-o3 | 16.46% | 23.20% | 3.35% | 23.13% | 24.49% | 38.89% | 11.75% | 9.23% | 18.18% | 7.41% | 25.34% | 13.92% | 9.23% |
| Qwen2.5-VL-72B-Instruct | 15.90% | 16.00% | 5.59% | 28.57% | 18.37% | 16.67% | 12.77% | 4.62% | 14.77% | 19.75% | 17.12% | 18.99% | 4.62% |
| Qwen2.5-VL-32B-Instruct | 12.76% | 16.00% | 3.35% | 19.05% | 22.45% | 5.56% | 9.72% | 3.08% | 10.23% | 14.81% | 15.75% | 16.46% | 3.08% |
| Qwen2.5-VL-7B-Instruct | 10.88% | 12.00% | 3.35% | 19.05% | 12.24% | 5.56% | 7.44% | 3.08% | 11.36% | 9.88% | 8.90% | 18.99% | 3.08% |
| InternVL2.5-78B-MPO | 11.92% | 9.60% | 6.70% | 19.73% | 18.37% | 27.78% | 7.44% | 4.62% | 7.95% | 13.58% | 11.64% | 22.78% | 4.62% |

## 2.5. Evaluation Strategy

Due to multi-image composition and our instance construction process, models' search behaviors can be sensitive to input perturbations. As a result, a single binary correctness score (0/1) is often insufficient to diagnose how a model succeeds or fails. We therefore report standard accuracy as the primary metric for ranking, and additionally introduce structure-aware diagnostic scores for multi-choice and sorting questions to expose partial progress signals such as set coverage and pairwise ordering consistency.

**Multi-Choice Questions.** For a multi-choice instance, the ground-truth answer $G$ and the model prediction $P$ are both subsets of the candidate set $C$. Binary correctness assigns 1 only when $P = G$, but it cannot distinguish predictions that are directionally correct yet incomplete from those that are dominated by distractors. To characterize structural consistency at the set level, we decompose prediction outcomes into:

$$TP = |P \cap G|, \quad FP = |P \setminus G|, \quad FN = |G \setminus P|. \quad (1)$$

Here, $TP$ counts correctly selected elements, $FN$ captures missing necessary evidence, and $FP$ reflects erroneously selected distractors. We further define set coverage:

$$\text{Coverage}(P, G) = \frac{|P \cap G|}{|G|}, \quad (2)$$

which measures how completely the model recovers the required evidence. Importantly, coverage is used for diagnostic analysis: it indicates whether the model is moving in a correct direction even when the final set prediction has not fully converged (*e.g.*, high $TP$ with non-zero $FN$), while $FP$ reveals susceptibility to irrelevant searched content.

**Sorting Questions.** In a sorting instance, the ground-truth $G$ and prediction $P$ define an ordering over the same $n$ elements. Exact-match accuracy is strict: a prediction must match the full ordering to be counted correct. In practice, models may capture many correct relative relations while failing to produce the exact final ranking. We therefore treat sorting as a set of pairwise relations and measure pairwise ordering consistency:

$$\text{Order}(P, G) = \frac{1}{\binom{n}{2}} \sum_{i<j} \mathbb{1}\left[\text{order}_P(i,j) = \text{order}_G(i,j)\right], \quad (3)$$

where $\text{order}_P(i,j)$ denotes the relative order between items $i$ and $j$ implied by $P$. This metric quantifies how well a model recovers the global relative structure even when the exact ordering is not perfectly matched.

# 3. Experiments

## 3.1. Settings

We systematically evaluate a diverse set of representative VLMs on NAVIGATE, covering mainstream open-source models, closed-source commercial models, and search-enabled agentic models. We evaluate each model in its native, released configuration without modifying architectures or inference procedures. For search-enabled models, we invoke their built-in search capabilities through the official APIs using default settings to reflect out-of-the-box behavior. Experiments run on a machine with 8 NVIDIA A100 GPUs, and closed-source models are queried via their official APIs under consistent calling protocols.

## 3.2. Main Results

In terms of overall accuracy, NAVIGATE poses a substantial challenge to existing models. As shown in Table 2, even the best-performing model, Gemini-3-Pro-Preview-Search, achieves an overall accuracy of only 36.4%. This result indicates that the multi-image, search-driven reasoning tasks characterized by NAVIGATE significantly exceed the capabilities of current VLMs.

Comparing model types shows that models equipped with search capabilities consistently outperform those without explicit search mechanisms, highlighting the critical role of external search in solving NAVIGATE tasks. Notably, this advantage persists even within the same model family. For instance, in the GPT-5 series, the search-enabled variant achieves an accuracy of 33.3%, compared to only 27.2% for its non-search counterpart. A similar trend is observed in the GPT-5.1 series. These results clearly demonstrate that the adoption of search strategies constitutes a key factor influencing model performance on NAVIGATE.

Notably, strong general-purpose multimodal capabilities or larger model scales do not necessarily translate into higher NAVIGATE performance: several large VLMs still achieve below 20% accuracy. This indicates a gap between generic vision–language understanding and the search-centric reasoning required by NAVIGATE, highlighting the limits of parametric knowledge and internal reasoning alone. Consequently, NAVIGATE provides a distinct testbed for evaluating search-driven, cross-image reasoning.

Overall, the performance variance across models indicates that NAVIGATE provides strong discriminative power. However, binary (0/1) accuracy alone remains insufficient to fully capture differences in models' search behaviors. Therefore, in the following analyses, we further investigate model behaviors from more fine-grained perspectives.

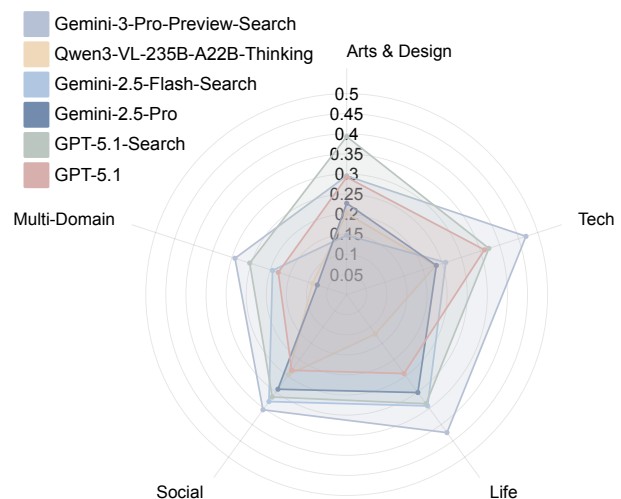

*Figure 4.* Sub-task performance analysis on NAVIGATE. Performance varies substantially by domain: Tech and Life are the most discriminative with the largest inter-model spread, Social exhibits comparatively tighter clustering, and Arts shows the greatest dispersion, indicating higher sensitivity to weakly structured cues.

## 3.3. Domain-Wise Analysis

From the single-domain breakdown (see Figure 4), models exhibit clear "strength–weakness" patterns across domains rather than uniform gains. Overall, Tech and Life are more discriminative: some models expand noticeably while others shrink, suggesting these questions rely more on structured interpretation of visual cues and effective integration of searched evidence, which amplifies performance gaps.

Concretely, Gemini-3-Pro-Preview-Search demonstrates the most balanced performance across domains, suggesting stable cross-domain generalization in single-domain settings. In contrast, several models appear more domain-skewed, *e.g.*, GPT-5.1-Search is particularly strong on Arts & Design, while its advantages are less pronounced on Tech / Life, reflecting differences in how search and reasoning strategies adapt to domain-specific cues.

Moreover, Social shows relatively smaller variance with closer curves across models, implying lower separability than Tech / Life, whereas Arts & Design is more dispersed, indicating that aesthetic or weakly-structured cues remain challenging and expose differences in evidence localization and filtering robustness.

## 3.4. Difficulty-Level Analysis

Comparing single-image and multi-image settings yields two consistent observations that reflect NAVIGATE's goal of diagnosing search decision-making.

**Single-Image *vs.* Multi-Image Gap.** Accuracy is substantially higher on single-image tasks than on multi-image

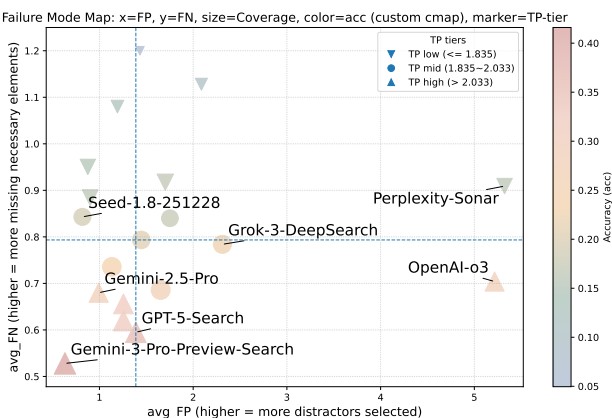

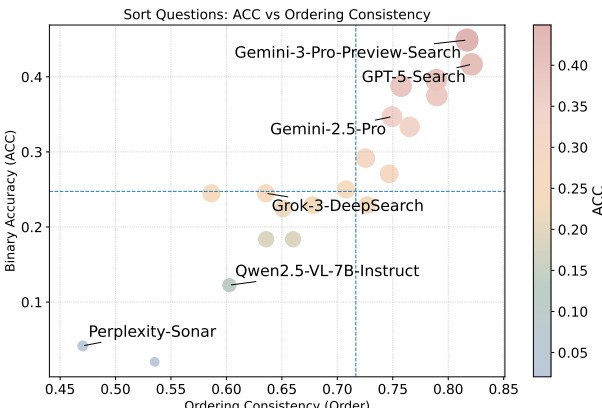

*Figure 5.* Set-level failure landscape for multi-choice questions (FP, FN, TP tiers, Coverage, Acc). Beyond accuracy, the $(TP, FP, FN)$ decomposition reveals whether failures stem from missing search (high $FN$) or from over-trusting irrelevant searching content (high $FP$), highlighting that the answer requires both targeted search and effective distractor suppression.

*Figure 6.* Accuracy and ordering consistency analysis on sorting questions. Ordering consistency largely tracks accuracy, but better diagnoses two search-linked failure modes: evidence-insufficient partial structure (high consistency despite low accuracy) versus misdirected strategy with widespread inversions (low consistency).

tasks. This gap is not merely due to increased visual complexity. Single-image questions are often solvable within a closed-evidence context. In contrast, multi-image tasks require establishing semantic relations across images and integrating information that is frequently distributed across multiple sources. As a result, models must identify missing evidence, invoke search, and aggregate retrieved content into outputs, which places greater demands on both search and integration capacity. A representative failure case caused by improper search is provided in the appendix A.2.

**Single-Image Tasks as a Closed-Evidence Baseline.** For single-image questions, the required evidence is typically contained within the image itself, making them solvable without searching. We therefore use non-search models as a baseline for this subset: their performance reflects the attainable accuracy when an agent consistently abstains from search. Notably, several search-enabled models underperform this baseline, suggesting that effective agents must not only search for information when needed, but also refrain from unnecessary search when the visual context is sufficient. Such over-search can introduce irrelevant or conflicting information and disrupt otherwise correct visual reasoning, making the single-image subset a controlled probe of searching restraint.

### 3.5. Set-Level Analysis of Multi-Choice Questions

For multi-choice questions, we use the set-level decomposition $(TP, FP, FN)$ (see Figure 5) to diagnose search behaviors beyond final accuracy. Here, $TP$ and Coverage indicate whether the model searches and aggregates the right evidence (where to search); $FN$ reflects insufficient search depth or incomplete aggregation; and $FP$ measures

susceptibility to irrelevant retrieved content, *i.e.*, whether the model can filter noise once search is invoked.

Overall, Gemini-3-Pro-Preview-Search achieves the best accuracy and exhibits consistently strong structural statistics (higher $TP$/Coverage with lower $FN$/$FP$), suggesting that it can both target relevant evidence and suppress distractors. In contrast, a representative failure mode is exemplified by Perplexity-Sonar: despite comparable $TP$ and Coverage to several models, it incurs substantially higher $FP$, indicating that search is triggered but poorly controlled, causing predictions to be dominated by distractors. More broadly, models tend to fall into two regimes: (i) high $FN$ with low $FP$, suggesting limited search depth and evidence omission; and (ii) high $FP$, indicating unstable search/filtering under irrelevant cues. These patterns show that multi-choice errors are not solely due to missing evidence, but often arise from misguided or over-trusting search, reinforcing NAVIGATE's focus on evaluating not only when to search, but also how to search and integrate evidence reliably.

### 3.6. Analysis of Sorting Questions

For sorting questions, binary accuracy and ordering consistency (Figure 6) show broadly aligned trends, but the latter better reveals failures beyond exact-match correctness. Even when rankings are imperfect, many models still recover many correct pairwise relations. This suggests models often infer plausible relative relations, yet fail to reach the globally correct ordering when critical evidence is missing or only partially searched—a failure mode tied to incomplete search. Notably, some models with low exact-match accuracy achieve comparatively high ordering consistency, indicating that they can form an approximate structure but cannot resolve the remaining global constraints without targeted search. Conversely, models with low ordering consis-

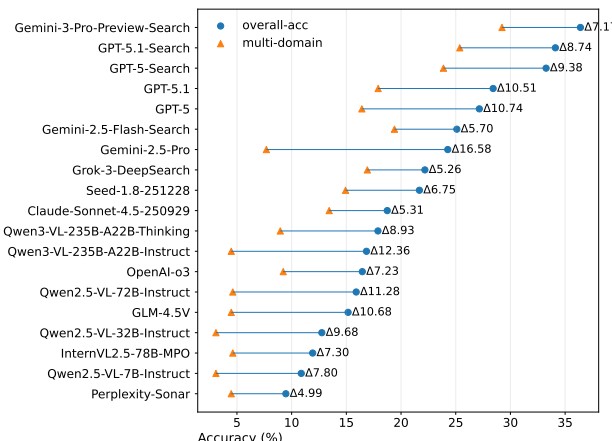

*Figure 7.* Overall accuracy *vs.* multi-domain accuracy across models. Multi-domain accuracy drops sharply for all models, while search-enabled systems maintain a consistent edge, highlighting search as critical for bridging cross-domain semantic gaps.

tency tend to exhibit broader structural inversions, reflecting misguided evidence use or incorrect ordering decisions.

Overall, ordering consistency complements accuracy by separating two search-relevant failure modes in sorting: (i) insufficient evidence (reasonable pairwise structure but incomplete convergence), and (ii) misdirected strategy (widespread inversions), reinforcing NAVIGATE's focus on evaluating whether agents can search and integrate the right evidence to form correct global structures.

### 3.7. Multi-Domain Tasks

To stress-test search decision-making and cross-image integration, NAVIGATE includes multi-domain tasks in which the input images originate from different domains and exhibit limited surface-level overlap. Solving these instances typically requires models to (i) recognize that external retrieval is necessary, (ii) formulate queries that explicitly bridge the semantic gap across images, and (iii) aggregate and reconcile evidence from multiple web sources to uncover the latent relation. As shown in Figure 7, all evaluated models suffer a substantial accuracy drop on this subset, suggesting that cross-domain relation discovery and multi-source evidence integration remain difficult for current VLMs. In contrast, search-enabled models consistently outperform their non-search counterparts, indicating that retrieval is important for expanding evidence coverage and linking otherwise disjoint visual cues. However, the remaining failures further imply that successful performance hinges not merely on invoking search, but on whether retrieval is correctly targeted (e.g., appropriate query refinement and source selection) and whether the returned evidence is integrated robustly without being distracted by irrelevant results.

## 4. Related Works

**Search-Augmented VLMs.** Recent VLMs (Bai et al., 2025b; Chen et al., 2024b; Comanici et al., 2025; Zheng et al., 2024a; Wang et al., 2025) have made rapid progress on multi-image understanding and visual question answering, demonstrating strong compositional reasoning and instruction following over rich visual inputs. However, their knowledge is still largely encoded in parametric memory, which makes it difficult to acquire up-to-date information or verify long-tail facts in open environments. To overcome this limitation, an increasing number of works augment models with external search, allowing them to query online sources and ground answers in retrieved evidence (Anthropic, 2025; ByteDance Seed, 2025; Chen et al., 2024a; Google Deep-Mind, 2025; He et al., 2024; OpenAI, 2025b; Perplexity Team, 2025; Team et al., 2025b; xAI, 2025; Team et al., 2025a). This direction pushes VLMs beyond static generation toward search-enabled multimodal agents that can retrieve, synthesize, and update information when solving complex real-world queries.

**Search & Browsing Benchmarks.** Early evaluation benchmarks (Chen et al., 2025a; Zheng et al., 2024b; Zhang et al., 2024; Yue et al., 2024; Wang et al.; Masry et al., 2022; Cheng et al., 2025; Fu et al., 2024; Jiang et al., 2025; Li et al., 2023; Liu et al., 2024a;b) mainly target closed-world settings, assessing perception and reasoning based on models' parametric knowledge. With the rapid emergence of web-search-enabled models, such benchmarks are insufficient to characterize open-environment performance. BrowseComp (Wei et al., 2025) and HLE (Phan et al., 2025) require web search to solve complex tasks and have inspired follow-up suites (Mialon et al., 2023; Zhou et al., 2025; Phan et al., 2025; Zheng et al., 2025; Yang et al., 2023; Wong et al., 2025; Chen et al., 2025b); however, they are largely text-only and do not reflect the multi-image content common in real applications. MM-BrowseComp (Li et al., 2025) incorporates visual inputs but focuses on relatively simple, single-point retrieval, limiting its coverage of searching and reasoning. LiveVQA (Fu et al., 2025) instead emphasizes time-sensitive news search, prioritizing information freshness over compositional reasoning.

Motivated by these limitations, we propose NAVIGATE to systematically evaluate models' search and reasoning in image-centric settings, offering a more comprehensive, application-oriented benchmark for VLMs.

## 5. Conclusion

Image-centric questions in open-web environments require models to infer search targets from visual cues and iteratively refine search trajectories, yet this setting is rarely evaluated in existing benchmarks. To support this evaluation, we

introduce NAVIGATE, a benchmark that emphasizes search decision-making. Experiments show that current systems remain struggling in these settings: even the best-performing Gemini-3-Pro-Preview-Search reaches only 36.4% accuracy. Errors frequently arise from a combination of imperfect multi-image grounding, incomplete cross-image evidence integration, and unstable path control during search. We hope NAVIGATE could serve as a diagnostic testbed that encourages progress toward more dependable visual search planning and iterative reasoning.

## Acknowledgements

This work is funded by the National Natural Science Foundation of China (Grant No. 62372223 and U24A20330), the Youth PhD Student Research Project under the National Natural Science Foundation (No. 623B2050), the Key R&D Program of Xinjiang Uygur Autonomous Region (Grant No. 2025B03043-1), Nanjing University-China Mobile Communications Group Co., Ltd. Joint Institute, and Fundamental and Interdisciplinary Disciplines Breakthrough Plan of the Ministry of Education of China (No. JYB2025XDXM118).

## Impact Statement

This paper presents work whose goal is to advance the field of machine learning. There are many potential societal consequences of our work, none of which we feel must be specifically highlighted here.

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

# A. Appendix

## A.1. Detailed Results for Multi-Choice Questions

This section presents the detailed statistics for multi-choice questions (see Figure 3), including the distributions of TP, FP, FN, and Coverage across different models. These results complement the main analysis by providing fine-grained quantitative evidence of models' information coverage, omission patterns, and susceptibility to irrelevant distractors. The corresponding figures visualize how different models balance search direction, search depth, and noise robustness in multi-choice settings.

Table 3. Set-level statistics ($TP_{avg}$, $FP_{avg}$, $FN_{avg}$, $Coverage_{avg}$) and accuracy for multi-choice questions.

| Model | $TP_{avg}$ | $FP_{avg}$ | $FN_{avg}$ | $Coverage_{avg}$ | Accuracy |
|---|---|---|---|---|---|
| **VLMs with Search** | | | | | |
| Gemini-3-Pro-Preview-Search | 2.22 | 0.63 | 0.53 | 0.80 | 41.60 |
| GPT-5-Search | 2.12 | 1.39 | 0.60 | 0.78 | 36.36 |
| GPT-5.1-Search | 2.10 | 1.26 | 0.62 | 0.78 | 32.23 |
| Gemini-2.5-Flash-Search | 2.10 | 1.26 | 0.66 | 0.76 | 32.23 |
| Grok-3-DeepSearch | 1.97 | 2.31 | 0.78 | 0.71 | 22.31 |
| Seed-1.8-251228 | 1.88 | 0.82 | 0.84 | 0.69 | 19.01 |
| GLM-4.5V | 1.83 | 0.90 | 0.88 | 0.66 | 16.00 |
| Perplexity-Sonar | 1.81 | 5.32 | 0.91 | 0.65 | 16.00 |
| Claude-Sonnet-4.5-250929 | 1.80 | 1.70 | 0.92 | 0.67 | 16.80 |
| **VLMs without Search** | | | | | |
| Gemini-2.5-Pro | 2.01 | 0.99 | 0.70 | 0.74 | 28.00 |
| OpenAI-o3 | 2.03 | 5.22 | 0.70 | 0.74 | 28.00 |
| GPT-5.1 | 1.93 | 1.65 | 0.70 | 0.74 | 27.27 |
| GPT-5 | 1.92 | 1.13 | 0.70 | 0.73 | 23.20 |
| Qwen3-VL-235B-A22B-Instruct | 1.92 | 1.45 | 0.80 | 0.70 | 20.66 |
| Qwen2.5-VL-32B-Instruct | 1.91 | 1.75 | 0.80 | 0.68 | 17.36 |
| Qwen3-VL-235B-A22B-Thinking | 1.76 | 0.88 | 1.00 | 0.65 | 14.88 |
| Qwen2.5-VL-72B-Instruct | 1.67 | 1.19 | 1.10 | 0.60 | 12.00 |
| Qwen2.5-VL-7B-Instruct | 1.62 | 2.09 | 1.10 | 0.59 | 9.60 |
| InternVL2.5-78B-MPO | 1.55 | 1.43 | 1.20 | 0.54 | 4.96 |

## A.2. Single-image Questions Error Case Analysis

Here we present a representative failure case (see Figure 8). The underlying question is a simple single-image problem that can be correctly answered using the model's internal knowledge without any external search. However, the model unnecessarily triggers search, retrieves irrelevant and distracting evidence, and then incorporates this noise into its final decision, leading to an incorrect answer. This error pattern highlights that search is not always beneficial and directly supports the importance of deciding when to search in practical settings.

## A.3. Failure-Driven Collection Paradigm

We do not restrict our candidate pool to only "extreme failures." Instead, failure-driven candidates are collected via pre-defined trigger rules reflecting search-related breakdowns, such as incorrect search triggering (search invoked when unnecessary, or omitted when required), misguided search direction (queries that fail to capture the visual intent or lead to irrelevant sources), and failures to filter or integrate retrieved information (e.g., being distracted by irrelevant results or adopting unsupported evidence). Each candidate is then double-checked by two annotators to confirm whether it constitutes a failure and to determine its failure type. To reduce overfitting to any specific tool or interaction pattern, we perform stratified sampling with quota control across different system categories and tool configurations, and we further apply clustering-based deduplication over both task texts and action trajectories. For traceability, we retain provenance and version metadata for every sample, including source category, tool type, time window (or simulation seed), and the annotation-guideline version, enabling end-to-end auditing of data generation and labeling.

### A.4. Example Questions from NAVIGATE

This section presents representative example questions from NAVIGATE to illustrate the data format and task characteristics (see Figure 9-12). The examples cover multi-choice questions, sorting questions, and cross-domain settings, reflecting the diverse search and reasoning scenarios encountered during evaluation. These examples are provided for illustrative purposes only, without introducing additional quantitative analysis or conclusions.

### A.5. Qualitative Examples of Model Outputs

We provide representative examples of model outputs for selected questions to qualitatively illustrate different success and failure patterns (see Figure 13&14). These examples highlight typical behaviors observed in the evaluation, such as correct search direction with incomplete information, excessive retrieval of irrelevant content, and failure to establish global consistency across multiple images.

### A.6. Evaluation Prompts

For each question type in NAVIGATE, we design a dedicated prompt tailored to the corresponding task format (see Figure 15). All evaluated models are tested using these fixed prompts without any modification or task-specific tuning, ensuring a fair and consistent evaluation setting across different models. During evaluation, models are required to follow the prompt instructions strictly, including the specified output format (and reasoning requirements when applicable). For open-ended questions, we employ GPT-4o as an automatic grader to assess answer correctness under a strict equivalence rubric (*e.g.*, accepting only semantically identical short answers such as standard aliases/synonyms or unit/format variants). If the grader produces an uncertain decision or conflicts with rule-based canonical matching, we fall back to manual adjudication under the same rubric.

To improve scoring consistency, we canonicalize both predictions and references into a normalized form (*e.g.*, case/punctuation/number formatting) before grading, and run the grader with deterministic decoding. Using GPT-4o as an automatic judge for *simple* open-ended answers has been widely adopted and validated in prior work and practice; nevertheless, we additionally validate grader reliability in our setting. On a random sample of 100 OEQ instances (Gemini-3-Pro-Preview-Search outputs), GPT-4o grading matches human scoring under our rubric. All prompts used for evaluation, including task prompts and grading prompts, are reported in full in this section to facilitate transparency and reproducibility.

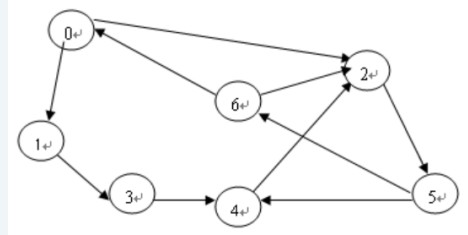

**Which of the following algorithms can be used to detect the special structure in <img1>?**
A. DFS;          B. Topological Sort;
C. BFS;          D. Kruskal's Algorithm

## GPT-5-Search

**Step-by-step reasoning with sources:**
- **The image shows a directed graph. Detecting a "special structure" in such contexts commonly refers to detecting cycles.**
- **DFS can detect cycles in directed graphs by finding back edges or using color-marking.**
- **Topological sort, such as Kahn's algorithm, can detect cycles in directed graphs: if the algorithm cannot process all vertices, the graph contains a cycle.**
- **BFS alone is not the standard approach for detecting cycles in directed graphs; typical BFS-based cycle detection is for undirected graphs.**
- **Kruskal's algorithm finds a minimum spanning tree in weighted undirected graphs and is not applicable for detecting cycles in directed graphs.**

**Conclusion: DFS and Topological Sort can be used to detect the special structure, namely cycles, in the directed graph. BFS and Kruskal's algorithm do not apply.**
**\boxed{AB}**

## GPT-5

**Step-by-step reasoning:**
- **The figure is a directed graph with cycles. Detecting the "special structure" here refers to detecting cycles.**
- **DFS can detect cycles in directed graphs by finding back edges during traversal.**
- **Topological sort, such as Kahn's algorithm, can detect a cycle when not all vertices can be processed.**
- **Although Kahn's algorithm is often implemented using a queue, this does not mean that generic BFS should be selected as a separate cycle-detection algorithm for directed graphs in this question.**
- **Kruskal's algorithm is designed for minimum spanning trees in undirected graphs. Its union-find cycle check does not apply to detecting cycles in directed graphs.**

**Final answer:**
**\boxed{ABC}**

*Figure 8.* A simple single-image question where unnecessary search retrieves irrelevant evidence and distracts the model, causing an incorrect prediction—highlighting the importance of deciding when to search.

## Politics

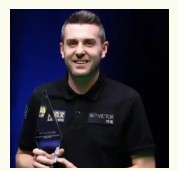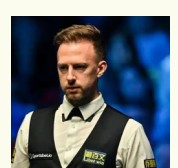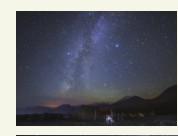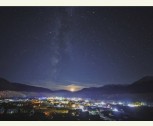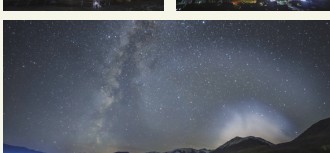

**Question**：**The following photo documents a well-known political event. May I ask how many countries' people were involved in this event?<img1><img2><img3><img4>**

**Answer**：**27**

## Economy

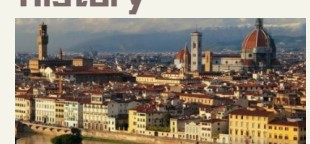

**Options:**
**A. Crude;**
**B.Cotton;** ✅
**C.Soybean;**
**D.Nickle**

**Question**：**This photo might be a picture of a futures contract for some commodity along with its trend line<img1>.**

**Answer**：**B**

## Sports

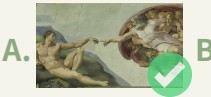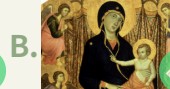

**Question**：**What was the greatest difference in scores between these two billiards players<img1><img2> in the final of the 2025 World Championship?**

**Answer**：**4**

## Astronomy

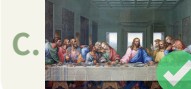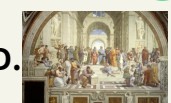

**Question**：**These photos were taken at different times in the same village. What is the village's altitude range?**

**Answer**：**1124-3000**

## History

**Question**：**Among the options, which ones reflect the ideology that emerged from the city in <img1>?**

**Options**：

A. ✅  B. ✅

C. ✅  D.

*Figure 9.* Data example from social domain.

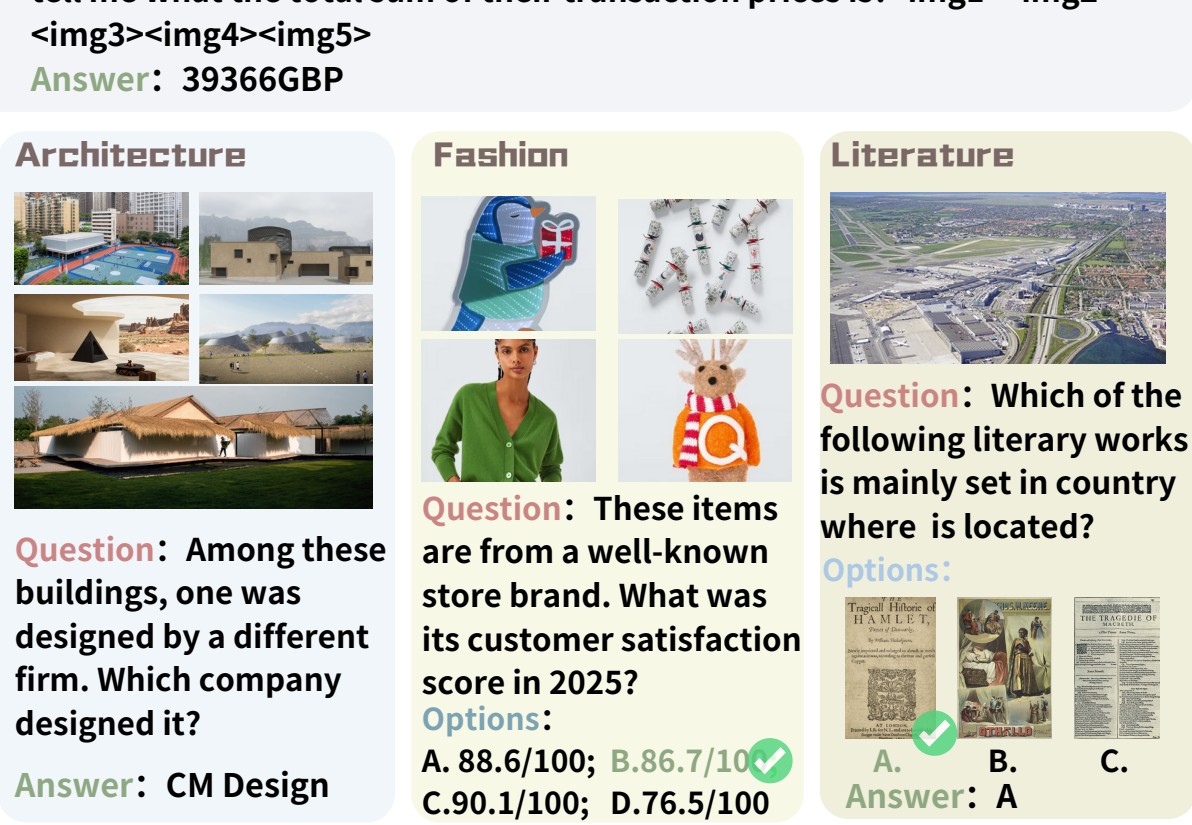

Figure 10. Data example from arts&design domain.

## Computer

**Question**: Which of the following companies and schools originated from the technological startups of <img1>? A.<img2>; B.<img3>; C.<img4>; D.<img5>; E.<img6>

A.     B.     C. ✅     D. ✅     E. ✅     F.

**Answer**: CDE

## Bio

**Question**: Among these three bird species, what is the altitude where the northernmost one was discovered?

**Options**:
A. 1636m;    B.1817m ✅
C.1933m;    D.212m

## Chemistry

**Question**: Please follow the correct GCLE production process in the correct order.

**Options**:

A.          B.

C.          D.

**Answer**: ACDB

## Physics

**Question**: The first stable antihydrogen confinement (≥1000 s) was achieved by which institution?Choose the only correct institution.

A.          B.

C. ✅        D.

*Figure 11.* Data example from technology and science domain.

**Game**
**Question**: The following are four different game scene pictures. Please tell me which game has the latest background time (please provide the game name and the start time of the story).

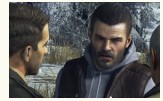 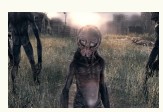
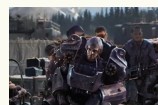 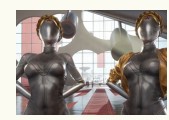

**Answer**: Fallout4/2077

**Life**
**Question**: If my primary requirement for buying a bag is its capacity, then which option of the bags should I choose?

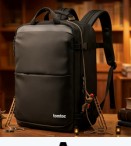
**A.**
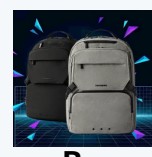
**B.**

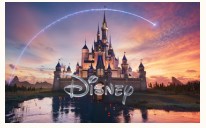
**C.**
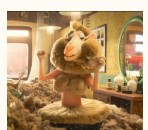
**D.**

**Comic**
**Question**: The following is the unique opening sequence of a certain animated film, along with one of characters. Could you please tell me what is the most popular song of the voice actor for this character?

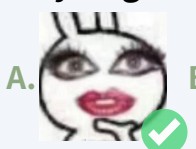 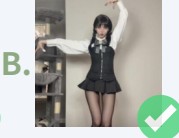

**Answer**: shape of you

**Music**
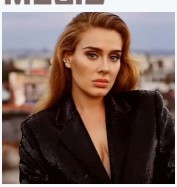 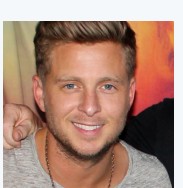

**Question**: What album did Figure One release after their last album with Figure Two?

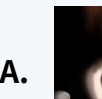 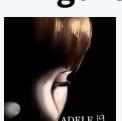
**A.** **B.**

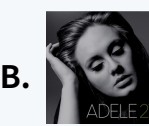
**C.** **D.**
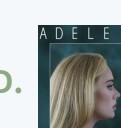

**Movie**
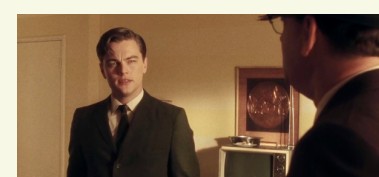

**Question**: Select the false identity assumed by the actor shown in the certain film.

**A.** 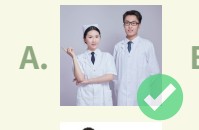 **B.** 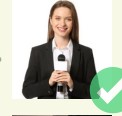

**C.** 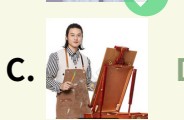 **D.** 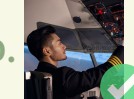

**Answer**: ABD

**Meme**
**Question**: Which of these meme pictures in options derived from Lady Gaga?

**A.** 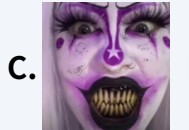 **B.** 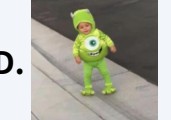

**C.** **D.**

**E.** 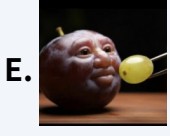

**Answer**: AB

*Figure 12.* Data example from daily-life domain.

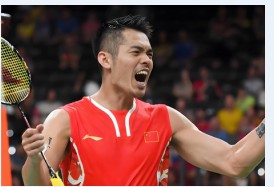 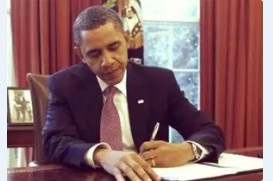 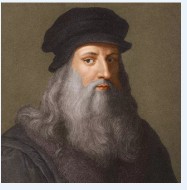

**Question**: From image 1-3 infer the share trait,and choose the one does not match is from the following options

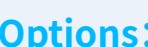
**Options:**

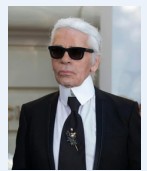 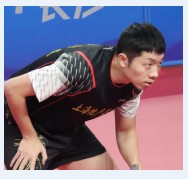 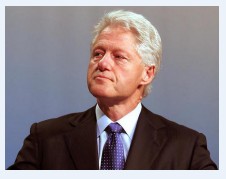 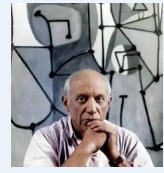

**Answer**: **A**

## Gemini-3-Pro-Preview-Search

The problem asks to select the option that does not match the share trait.

*Step 1: Identify the subject of the question pictures.*
fig1: **Lin Dan**, a Chinese badminton player.
fig2: **Barack Obama,** the 44th U.S. President.
fig3: **Leonardo da Vinci**, the Renaissance polymath (portrait/engraving).

*Step 2: Identify the subjects of the options.*
A: **Karl Lagerfeld**, the renowned German fashion designer and creative director of Chanel.
B: **Xu Xin**, a Chinese table tennis player.
C: **Bill Clinton**, the 42nd U.S. President.
D: **Pablo Picasso**, the Spanish painter.
Step 3: Determine the common characteristic linking the main picture and the options.**

*Analyzing the subjects, a specific trait connects* most of them: Left-handedness.
Xu Xin is a famous left-handed table tennis player (plays with his left hand).
Bill Clinton is a famous left-handed president.
Karl Lagerfeld is also frequently cited in lists of famous left-handed people (and often wore a glove on his left hand).
*Step 4: Ideåntify the outlier.*
Pablo Picasso (Option D) historical evidence and photographs show that he was right-handed (he painted and wrote with his right hand).
Since the main picture and options A,B, C all represent famous left-handed individuals, Pablo Picasso (who was right-handed) is the one who does not match the category established by the picture.
\boxed{D}

## GPT-5.1-Search

[Reasoning Summary]
- Observed which hand each person uses for their main activity (writing, playing, etc.) in images 1–3.
- Recognized that the three people in images 1–3 are widely documented as left-handed individuals.
- Checked web sources listing a well-known fashion designer (image4) among famous left-handers.
- Verified that the table-tennis professional corresponding to image5 has an officially listed playing style as right-handed.
- Confirmed that the statesman in image6 is documented as left-handed.
- Could not find strong evidence that the artist in image7 is right-handed; the puzzle most clearly contrasts image5 with the others based on handedness.

[Final Answer]
The shared trait in images 1–3 is **being left-handed**.
Among the options, **B (image5)** shows a person who is **right-handed**, so **option B does NOT match**.[M. F. Husain - Wikipedia](Url-1)[MF Husain was forced into exile; now his work finds permanent home in ...](url-2)  [M.F. Husain | Artist, Modernist, Painter, Printmaker, Life ...](url-3)

*Figure 13.* Example outputs from Gemini-3-Pro-Preview-Search and GPT-5.1-Search.

## seed-1-8-251228

- First, identify the status of each person in images 1–3: - Image 1 (Lin Dan): Retired professional badminton player, no longer active in competitive sports.
  - Image 2 (Barack Obama): Retired former US president, no longer serving in political office.
  - Image 3 (Leonardo da Vinci): Deceased, no longer active in his creative/scientific work.
- The shared pattern is: All individuals in images 1–3 are no longer active in their primary professional role, either retired or deceased.
- Now evaluate each option:
  - Option A (Karl Lagerfeld): Deceased fashion designer, no longer active.
  - Option B (Xu Xin): Active professional table tennis player, currently competing in matches as shown in the image.
  - Option C (Bill Clinton): Retired former US president, no longer active in political office.
  - Option D (Pablo Picasso): Deceased artist, no longer active.
- Option B is the only one that does not fit the shared pattern, as the individual is still active in their primary profession.

## GPT-5-Search

- Observed that in image 1 the athlete clearly holds the racket in the left hand, implying left-hand dominance.
- Recognized that the figures suggested by images 2 and 3 are widely known examples of famous left-handers (one a political leader, one an artist).
- Thus inferred that the shared trait across images 1–3 is "being left-handed."
- Checked that the prominent badminton player suggested by image 1 is indeed a left-handed player, and that the famous table-tennis player corresponding to image 5 is a right-handed player.
- In the options, image 5 uniquely shows a racket held in the right hand, while the people implied by the other options are commonly cited as left-handers or associated with left-handedness.
- Therefore, option B does not share the left-handed trait and is the odd one out.

[Final Answer]
The shared trait of images 1–3 is that they depict famous left-handed people. Among the options, image 5 (Option **B**) shows a right-handed person and therefore does **not** match.

*Figure 14.* Example outputs from Seed-1-8-251228 and GPT-5-Search.

## Query prompt：

**"multi-choice":** (The following is a multiple-choice question, which means that there may be more than one correct answer. You should consider retrieving up-to-date, authoritative information if necessary, but it is up to you to decide whether external search is needed. After gathering sufficient evidence, analyze the question and all options carefully, selecting all and only the correct options while avoiding incorrect ones. Please reason step by step, and put your final answer within \boxed{}. You only need to give the answer options such as \boxed{ABCD}.)

**"sort":** (The following is a sorting question, which means that the answer must include all options in a specific order. You may conduct external web search to gather factual, historical, or contextual information relevant to each option if necessary, but the decision to do so is left to you. Do NOT determine the order based only on intuition when reliable evidence is required. After gathering sufficient information, analyze the relationships, constraints, and comparison criteria among the options to determine the most accurate ordering. Please reason step by step, and put your final answer within \boxed{}. You only need to give the ordered options such as \boxed{ABCD}.)

**"single-choice":** (The following is a single-choice question, which means that there is only one correct answer. You may perform external web search to verify facts, resolve ambiguities, or eliminate incorrect options when needed, but you are not required to do so in all cases. After obtaining sufficient evidence, analyze the problem step by step and select the unique correct answer. Put your final answer within \boxed{}. You only need to give the answer option such as \boxed{B}.)

**"open-ended":** (The following is an open-ended question with no answer options provided. You may conduct external web search to identify relevant entities, events, concepts, background knowledge, or real-world facts implied by the question and the images if necessary. After gathering and cross-checking sufficient information, perform careful reasoning to derive the most accurate answer. The final answer should be concise and should not exceed four words. Put your final answer within \boxed{}. You only need to give the pure answer such as \boxed{Kill Bill}.)

**"other":**(The following is a single-choice question, which means that there is only one correct answer. You may perform external web search to verify facts, resolve ambiguities, or eliminate incorrect options when needed, but you are not required to do so in all cases. After obtaining sufficient evidence, analyze the problem step by step and select the unique correct answer. Put your final answer within \boxed{}. You only need to give the answer option such as \boxed{B}.)

## Eval prompt：

You are a strict evaluator whose only job is to extract the final answer from a model's output according to the question type.

You will be given: A question、A question type (one of: multi-choice, multi-choices, single-choice, sort, open-end, other)、A model-generated answer (which may contain \boxed{} segments)

Notes:
- The answer may contain multiple \boxed{} segments, or none.
- Some \boxed{} segments may include extra text (e.g., 'A. apple').
- The answer may mix reasoning and final prediction.

Your tasks: Identify the final answer based on the question type、Extract the clean final answer only、Output a single string with no explanation.

Rules for different question types:

**single-choice:** There is exactly one correct option. Extract the final selected option (A/B/C/D…). Return only that letter, e.g., B.

**multi-choice/multi-choices:** There are multiple correct options. Collect all chosen options in the model's final answer. Output concatenated uppercase letters, e.g., ACD.

**sort:** The answer must include all options and their order matters. Extract the order exactly as predicted. Output a sequence like BDAC.

**open-end:** There are no options. Extract the final textual answer from inside \boxed{}. The answer should be no more than four words. Return pure text, e.g., Kill Bill.

**Judge:**You are a strict evaluator. Determine whether the predicted answer is semantically equivalent to the ground truth. Only reply with a single character: '1' if equivalent, '0' if not. Do NOT add explanations or any other text.Ground truth answer: {gt_answer}、Predicted answer: {pred_answer}、Reply with exactly 1 or 0.

*Figure 15.* Query prompt and eval prompt.

