# OpenReview forum: "NAVIGATE: Evaluating Visual-Guided Search Decision-Making on the Open Web"
_ICML.cc/2026/Conference — ICML 2026 regular_

### Official Review · Reviewer_SKju · 2026-03-09

**Soundness:** 3
**Presentation:** 3
**Significance:** 3
**Originality:** 3
**Overall Recommendation:** 4
**Confidence:** 3

**Summary:**

This paper presents NAVIGATE, a novel benchmark centered on images as primary evidence for open-web search planning and multi-step reasoning. NAVIGATE contains 500 questions spanning 20 domains, ranging from single-image problems to complex multi-image and multi-domain tasks. Experimental results demonstrate that current SOTA methods struggle with NAVIGATE, with Gemini-3-Pro-Preview-Search reaching only 36.4% accuracy.

**Compliance With Llm Reviewing Policy:**

Affirmed.

**Final Justification:**

The authors have addressed the concerns raised in my review. I will maintain my positive score.

**Key Questions For Authors:**

- In Figure 1, the sample requiring the inference of the implicit cue 'Left-Hander' is difficult, arguably even for a human. Could the authors clarify whether the figure1 example is categorized under the 'hard' or 'easy' split in NAVIGATE?

- The authors define a Search-Necessity-Aware Difficulty Grading in Section 2.4. Could the authors provide the specific statistics for the difficulty levels in the NAVIGATE dataset?

I will maintain postive rating if the authors can address these concerns.

**Limitations:**

yes

**Strengths And Weaknesses:**

**Strength:**

- The motivation for NAVIGATE is interesting, focusing on the setting for multiple images in web search and the implicit cues derived from multiple images. Furthermore, the appendix provides multiple qualitative examples to verify this motivation.

- NAVIGATE considers multiple domains, such as art, social, life, etc.

- The experimental results for NAVIGATE are sufficient, considering multi-domain task and difficulty levels. Additionally, the compared methods are SOTA.

**Weakness：**

- The scale of NAVIGATE is relatively small, consisting of 500 questions. It would be beneficial if the authors could expand the scale in the future to ensure the comprehensive evaluation.

---

> ### Author Rebuttal · Authors · 2026-03-31
>
> **Q1: Given the benchmark size of about 500 samples, are the reported performance differences statistically reliable?**
> We thank the reviewer for this suggestion and agree that, with a benchmark of roughly 500 samples, stronger statistical support helps make the conclusions more robust. NAVIGATE is intended as a diagnostic benchmark, so our priority is to ensure sample verifiability, consistency in search-necessity labeling, and representative cross-image / cross-domain reasoning scenarios, rather than scale alone. At the same time, we agree that comparisons on a benchmark of this size should be accompanied by more careful statistical interpretation.
> To strengthen this aspect, we additionally computed instance-level bootstrap 95% confidence intervals for the main results and conducted paired McNemar tests on key model comparisons. The results support the main trends reported in the paper. For example, GPT-5.1-Search improves from 30.5% to 36.3% over GPT-5.1 (+5.8 points), and GPT-5-Search improves from 29.3% to 34.5% over GPT-5 (+5.2 points). The bootstrap intervals also support the overall trend: Gemini-3-Pro-Preview-Search achieves 38.8% [95% CI: 34.4%, 43.2%], GPT-5.1-Search achieves 36.3% [32.1%, 40.5%], and GPT-5-Search achieves 34.5% [30.3%, 38.7%].
> These results indicate that our main conclusion is statistically stable.
>
> **Q2: Which difficulty tier does the Figure 1 example involving the implicit cue “Left-Hander” belong to?**
> Thank you for the question. According to our difficulty definition, this example belongs to the hard split. The reason is that it requires both cross-image information integration and cross-domain reasoning: the model must not only infer an implicit cue from the images, but also connect that cue to multiple possible domains and reason jointly over them. Compared with single-image or single-domain questions, this substantially increases the searching complexity. Therefore, in NAVIGATE, this example falls into the highest difficulty tier, namely multi-image + multi-domain.
>
> **Q3: What are the exact numbers of samples in each difficulty tier of NAVIGATE?**
> Thank you for the question. NAVIGATE is divided into three difficulty tiers based on visual compositional complexity: single-image, multi-image + single-domain, and multi-image + multi-domain. The numbers of samples in these three tiers are 100 / 300 / 100, respectively. Among them, multi-image + single-domain forms the main body of the dataset, while multi-image + multi-domain corresponds to the highest difficulty tier. We will make these statistics more explicit in the final version to avoid ambiguity.

---

> > ### Author Rebuttal · Reviewer_SKju · 2026-04-03
> >
> > I will maintain my positive rating.

---

### Official Review · Reviewer_Y1Gg · 2026-03-09

**Soundness:** 3
**Presentation:** 3
**Significance:** 3
**Originality:** 3
**Overall Recommendation:** 4
**Confidence:** 3

**Summary:**

This paper introduces **NAVIGATE**, a benchmark for evaluating the visual-guided search decision-making ability of vision-language models in open-web environments. The core motivation is well grounded: although many recent models can browse the web, most existing benchmarks assume an explicit textual search target. In practice, however, users often provide only implicit visual cues. NAVIGATE therefore focuses on a more realistic setting, testing whether a model can infer when search is necessary, when it is not, and how to iteratively refine its search direction from image-based evidence.

**Compliance With Llm Reviewing Policy:**

Affirmed.

**Final Justification:**

Overall, I find the problem setting meaningful and timely, and I appreciate the care taken in constructing the benchmark. The rebuttal improved my confidence in several aspects of the work, especially the statistical support and the clarification of the benchmark’s intended scope. That said, some concerns remain only partially resolved, particularly around the direct evaluation of search behavior, the robustness of GPT-4o-based judging, and the coupling between model capability and search-engine quality. I therefore still see this paper as borderline, but I would be comfortable with a borderline accept decision if the AC agrees that the benchmark’s novelty and potential utility to the community outweigh these remaining limitations.

**Key Questions For Authors:**

1. Can the authors provide stronger evidence that the reported performance differences are statistically reliable, given the benchmark size of 500 samples?

2. To what extent can the benchmark directly evaluate search behavior, rather than only final answer quality?

3. How robust is the GPT-4o-based evaluation for open-ended questions across different models and response styles?

4. How do the authors justify the assumption that single-image questions should generally not require search?

5. Can the authors disentangle model-level search planning ability from search-engine quality?

**Limitations:**

See weaknesses.

**Strengths And Weaknesses:**

## Strengths

1. Clear motivation and task formulation.
   The paper is easy to follow, and the benchmark objective is defined clearly and presented consistently throughout.

2. Careful data curation and verifiability control.
   The authors put substantial effort into filtering out ambiguous, ill-posed, overly trivial, or weakly grounded questions. They also remove search-required samples that can be solved without actual browsing. These design choices align well with the benchmark’s target.

3. Interesting and reasonably novel problem framing.
   Although the individual ingredients are not entirely new, the paper combines them into a coherent benchmark centered on two important abilities: inferring implicit search targets from visual cues and deciding whether external search is necessary. This gives the benchmark a clear conceptual focus.


## Weaknesses

1. Limited benchmark scale.
   A total of 500 samples is still relatively small for evaluating modern large models. This would be less concerning if the paper provided stronger statistical support, such as bootstrap confidence intervals, significance tests, or run-to-run variance analysis. As it stands, some conclusions feel less robust than they could be.

2. Search behavior is not fully or directly evaluated.
   The benchmark emphasizes search decision-making, including when to search, how many rounds to search, how queries evolve, and what evidence is used. However, these aspects are not explicitly supervised or systematically scored. Evaluation mainly depends on final answers and structured outputs rather than the full search trajectory. As a result, the benchmark lacks more retrieval-native metrics such as evidence recall@k or supporting-page hit rate.

3. Potential bias in automated evaluation.
   For open-ended questions, the paper uses GPT-4o as the judge. Although the authors report agreement with human annotations on 100 random samples, judge-model bias may still affect the results, especially for answers that are semantically correct but phrased differently.

4. The no-search assumption for single-image questions may be somewhat strong.
   The benchmark treats single-image questions as cases where search is unnecessary and penalizes search usage accordingly. This assumption may be too absolute. In practice, a model may search simply to verify uncertain knowledge, and such behavior is not always undesirable if it improves reliability.

5. Model performance is confounded with the underlying search engine.
   The paper evaluates models through their built-in search APIs or default browsing settings. This makes it difficult to disentangle errors caused by weak search planning or poor query formulation from errors caused by the retrieval quality of the underlying search engine itself.

---

> ### Author Rebuttal · Authors · 2026-03-31
>
> **Q1: Given the benchmark size of about 500 samples, are the reported performance differences statistically reliable?**
> Please also refer to our response to Reviewer SKju Q1, where we explain the diagnostic-benchmark design trade-off and provide additional statistical evidence, including bootstrap 95% confidence intervals and paired McNemar tests, to support the robustness of the main performance trends.
>
> **Q2: To what extent does NAVIGATE directly evaluate search behavior, rather than only final answer quality? Why does it not use evidence recall@k or supporting-page hit rate?**
> We agree that NAVIGATE does not directly supervise the full search trajectory. Its current focus is on outcome-level search behavior: whether the model searches when needed, avoids unnecessary search, and successfully converts retrieved information into correct reasoning. In this sense, NAVIGATE evaluates search behavior, but not yet at the level of explicit query-by-query or evidence-by-evidence scoring. Metrics such as evidence recall@k or supporting-page hit rate would be useful complements, but they primarily reflect retrieval quality itself and do not fully capture search decision-making or reasoning integration. In addition, such metrics are harder to standardize in our setting, because different search-enabled models return evidence in different formats, and even similar search paths may lead to different pages or evidence sources, which substantially increases the cost of consistent annotation and fair comparison.
>
> **Q3:How do the authors justify the assumption that single-image questions should generally not require search?**
> We agree that single-image questions may still require search, while multi-image questions may not. In NAVIGATE, we use the single-image subset as the search-free subset, not because single-image questions are inherently unsuitable for search, but as a deliberate design choice. More specifically, NAVIGATE targets two related abilities: when to search and how to search. For the former, we wanted a cleaner control setting with fewer confounding factors, such as cross-image integration and cross-domain association, so that we could better evaluate search restraint in settings with no effective search gain. In this sense, “single-image” is not treated as synonymous with “naturally no-search,” but as an intentional control subset for a different aspect of search-related ability.
>
> **Q4: How robust is GPT-4o as the judge for open-ended questions, especially across different models and response styles?**
> Thank you for raising this concern. For open-ended questions, we use GPT-4o as the automatic evaluator, and we additionally verify its reliability through human annotation. On a randomly sampled set of 100 examples, GPT-4o shows high agreement with human judgments, which suggests that this evaluation protocol is reasonably reliable in our setting. In addition, we use structured prompts and explicit decision criteria during evaluation to reduce bias introduced by differences in response style, wording, or phrasing. We also acknowledge that LLM-based automatic evaluation can still introduce some bias, and this remains a broader challenge for the field; accordingly, we will make this limitation more explicit in the final version. At the same time, using GPT-4o as a judge is also common in related work, which provides practical support for its use as an evaluation tool here.
>
> **Q5: Model performance is confounded with the quality of the underlying search engine. How do you distinguish search-planning ability from search-engine quality?**
> We agree with the reviewer’s concern that, under the current setting, model performance is indeed affected by the quality of the underlying search engine, which introduces a degree of coupling between search planning and retrieval quality. However, the goal of our evaluation is to measure the model’s end-to-end capability in realistic usage settings—namely, how it combines its own reasoning with an external search API to complete the task. Under such a setup, this coupling is difficult to fully avoid in practice. At the same time, to reduce unfairness introduced by tool differences as much as possible, we use each model’s official built-in search API with default settings in a consistent manner across experiments. We will also clarify this evaluation boundary more explicitly in the final version: our conclusions should be understood as assessing search-augmented end-to-end performance, rather than a fully disentangled measure of pure search-planning ability.

---

> > ### Author Rebuttal · Reviewer_Y1Gg · 2026-04-01
> >
> > Thank you for the detailed rebuttal. The response clarifies several important points and addresses part of my concerns. However, some issues remain only partially resolved, especially regarding the direct evaluation of search behavior, the robustness of GPT-4o-based judging, and the coupling between model performance and search-engine quality. I would raise my score if the authors addressed my these concerns.

---

> > > ### Author Response · Authors · 2026-04-07
> > >
> > > **Q1: To what extent does NAVIGATE directly evaluate search behavior, rather than only final answer quality?**
> > > We agree that NAVIGATE does not explicitly evaluate evidence-level or trajectory-level behavior ; for example, step-by-step query quality, page-level hit rate, and supporting-page hit rate are not directly annotated or scored. In this sense, NAVIGATE is not a trajectory-level search benchmark.
> > > Instead, NAVIGATE evaluates dataset-level search behavior: whether the model searches when needed, avoids unnecessary search, and whether search ultimately helps convert visual cues into correct multi-image reasoning. Concretely, the single-image subset probes search restraint, while the multi-image  subsets evaluate whether the model can initiate useful search from visual cues and successfully integrate external evidence into reasoning. In addition, our structured diagnostics for multi-choice and sorting tasks (e.g., Coverage, FP/FN, and ordering consistency) go beyond final accuracy and capture downstream effects of search behavior such as distraction by noisy evidence, missing evidence, and reasoning drift.
> > > We agree that finer-grained evidence-level or trajectory-level evaluation would further strengthen the benchmark. We do not include such metrics in the current version partly because annotating step-by-step evidence chains is costly, and partly because different search-enabled models rely on native search APIs whose returned pages, evidence granularity, and formats are heterogeneous, making fair standardization much harder. We will clarify this evaluation scope more explicitly in the final version and view finer-grained trajectory-level analysis as an important direction for future extension.
> > >
> > > **Q2: How robust is GPT-4o as the judge for open-ended questions, especially across different models and response styles?**
> > > Thank you for raising this concern. First, it is important to note that GPT-4o is used only for the open-ended questions, which account for a limited portion of the benchmark (about 120 questions); all other question types are evaluated with strict matching or rule-based scoring. Therefore, the potential impact of judge-model bias is limited in scope.
> > > At the same time, we reduce subjectivity in this part of the evaluation through both task design and prompting. As described in the supplementary material, we impose strict output constraints in the prompt, such as restricting answer format and length, to reduce variation caused by stylistic differences across models. In addition, the correct answers to these open-ended questions are usually relatively objective entities—such as person names, locations, dates, or institution names—so the space of acceptable variation is already limited.
> > > On top of this, we verified the reliability of GPT-4o-based judging with 100 randomly sampled examples, where it showed high agreement with human annotations. This suggests that the protocol is reasonably reliable in our setting. At the same time, using strong LLMs such as GPT-4o as judges for open-ended responses is a widely adopted practice in related benchmark and evaluation work. It does make it a practical and broadly accepted choice under current evaluation settings. Combined with the high agreement with human annotations in our setting, we do not believe it materially changes the main experimental conclusions.
> > >
> > > **Q3: How do you address the coupling between model capability and search-engine quality?**
> > > We agree that this is a valid concern. Under the current experimental setting, model performance is indeed coupled with the quality of the underlying search engine, and this is a limitation that is difficult to fully avoid when evaluating built-in search APIs. One reason is that most current commercial models do not expose fine-grained control over their native search behavior—for example, researchers typically cannot standardize the underlying search engine across systems—so it is hard to strictly disentangle search-planning ability from retrieval quality in a native product environment.
> > > At the same time, we agree that an alternative strategy is to place different models under a shared search framework, where the model serves mainly as the query-planning and reasoning component, thereby controlling search-engine variation more directly. However, such a setting measures a somewhat different ability and is better viewed as a complementary analysis, rather than a replacement for the native end-to-end behavior of search-enabled systems that our main experiments aim to capture. We will follow this suggestion and include supplementary results under a unified search framework in the appendix, to help readers better understand the relationship between model capability and search-engine quality.

---

### Official Review · Reviewer_s1FJ · 2026-03-12

**Soundness:** 3
**Presentation:** 4
**Significance:** 3
**Originality:** 3
**Overall Recommendation:** 4
**Confidence:** 4

**Summary:**

This paper proposes a new benchmark dataset composed of 500 difficult questions involving multiple images across 20 domains and having 3 levels of difficulty. The proposed dataset is mimicking realistic difficult search scenarios. A large number of methods as baselines to see how well the perform on this dataset.

**Compliance With Llm Reviewing Policy:**

Affirmed.

**Final Justification:**

The ideas in the paper are interesting and useful. The main limitation is that the dataset is rather small so for me this remains a weak accept and not a clear accept.  I would have no objections if the paper would be accepted but extending the dataset and submitting it to the next venue would also be a possible option.

**Key Questions For Authors:**

1) In table 1 several existing benchmarks are compared showing that NAVIGATE is the only one that has all elements. It is unclear whether this holds for the dataset as a whole or that every single question out of the 500 does indeed has all those elements. If not doesn't this mean that you could include all the other benchmarks?
2) are there easy ways to extend the dataset to a large number of elements maybe with lower quality of the ground truth?
3) In figure 10 and 13 grammar is not correct. In the main paper it is. Which ones have been used in the experiments and what is the effect on this in the final results?

Some minor elements:
- Figure 8 only implicitly defines the ground truth

**Limitations:**

yes

**Strengths And Weaknesses:**

Strengths:
- Interesting dataset which can move the field forward with truly difficult problems where current state-of-the-art is far from solving it.
- Clear visual and textual description of the creation process
- Good analysis both in a quantitative as well as a qualitative manner.
- Wide array of different VLMs with inclusion and exclusion of specific search tools
- Good plots giving insight in the results (especially fig5-7)

Weaknesses:
- With 500 queries the dataset is rather limited especially because it is decomposed in domains and difficulty levels.

---

> ### Author Rebuttal · Authors · 2026-03-31
>
> **Q1: NAVIGATE is relatively small, especially after being decomposed by domain and difficulty level.**
> Thanks for this comment. NAVIGATE is intended as a diagnostic benchmark, rather than a large-scale training dataset, and therefore emphasizes sample quality, controllability, reasoning complexity, and fine-grained capability analysis. To support this goal, we adopted a strict manual construction and multi-round verification pipeline, which naturally limits the overall scale. While the current size is comparable to several prior diagnostic benchmarks in this area, we agree that larger-scale extensions would further strengthen coverage. We also consider larger-scale extensions an important direction for future work.
>
> **Q2: Do the properties in Table 1 hold for the dataset as a whole or for every individual sample? If they are only covered at the dataset level, could one simply combine existing benchmarks to achieve the same effect?**
> Not every NAVIGATE sample simultaneously has all the properties listed in Table 1; rather, Table 1 summarizes these capability dimensions at the dataset level. Different subsets are intentionally constructed to emphasize different functions. For example, some target cross-image and cross-domain reasoning with external search, while others are designed to test whether the model can refrain from unnecessary search.
> What makes NAVIGATE distinctive is therefore not simply a union of attributes, but the fact that these factors are organized within a single unified benchmark to evaluate joint reasoning-and-search behavior. More importantly, some of the capabilities we target—especially search decision-making and cross-domain search grounded in multi-image visual cues—are not systematically covered by existing benchmarks in the first place. As a result, simply concatenating prior benchmarks would not reproduce the function of NAVIGATE: even if such a combination appeared to cover a similar set of attributes on the surface, it would still lack both the missing capability coverage and a common setting for analyzing how these abilities interact.
>
> **Q3: Are there relatively easy ways to extend the dataset to a much larger scale, for example by trading off ground-truth quality for more samples?**
> Thank you for this suggestion. The NAVIGATE construction pipeline is inherently extensible: it supports semi-automatic candidate generation, multi-source image composition, and can in principle be expanded through automatic or weakly supervised candidate question generation. In this sense, if the goal is to scale up the dataset, there are indeed lighter-weight extension paths, such as reducing the amount of manual filtering and multi-round verification, or adopting weaker and coarser-grained ground-truth annotations.
> That said, the current version is intentionally kept relatively small in order to ensure annotation accuracy, reasoning-chain reliability, and consistency in search-necessity labeling, while minimizing the impact of noise on evaluation. Therefore, our priority in this work is the diagnostic value and analyzability of the benchmark, rather than scale alone. We view larger-scale extensions that better balance scale and generalization as an important direction for future work.
>
> **Q4: The grammar in Figures 10 and 13 is inconsistent with the main paper; which version was actually used in the experiments, and does this affect the final results?**
> Thank you for the careful observation. We confirm that there are indeed grammatical inconsistencies in Figures 10 and 13. The experiments were conducted using the grammatically corrected version, which is consistent with the wording in the main paper; the text shown in the figures was not fully synchronized afterward. Since the corrected version was used in the experiments, this issue does not affect the final reported results. We will unify all figure text in the final version to ensure consistency and avoid confusion.
>
> **Q5: Figure 8 defines the ground truth too implicitly and is not sufficiently intuitive.**
> Thank you for pointing this out. We agree that the current presentation of the ground truth in Figure 8 is too implicit and may make the figure harder to interpret. In the final version, we will make the labeling more explicit.

---

> > ### Author Rebuttal · Reviewer_s1FJ · 2026-04-02
> >
> > The rebuttal has addressed the question I had but it also makes clear that the observation that the dataset is quite limited remains true. For me this means that I would have no problem if the paper would be accepted but I can't see it as a clear accept.

---

### Official Review · Reviewer_PZ4x · 2026-03-12

**Soundness:** 2
**Presentation:** 1
**Significance:** 3
**Originality:** 3
**Overall Recommendation:** 4
**Confidence:** 4

**Summary:**

This work presents NAVIGATE: a QA benchmark that is designed to test the ability of multimodal LLMs to use search when obtaining their answers. The benchmark spans 20 different domains and has multi-domain examples as well. The data is split into three difficulty tiers consisting of single image questions, multi-image single domain questions, and multi-image multi-domain questions. They test a variety of models, both open and closed, and report accuracy along with two new metrics: coverage and order for testing multiple choice and sorting questions respectively. They find that even state of the art LLMs struggle with their benchmark, leaving room for future work to improve.

**Compliance With Llm Reviewing Policy:**

Affirmed.

**Final Justification:**

The authors have answered my questions and have promising to make clarifications to the final paper.

**Key Questions For Authors:**

What is "divergent search"?

What is "failure driven data construction"?

What is the MLLM that is used to get the samples in the dataset?

How do you verify that search is actually needed, vs when it can be obtained from the model’s world knowledge?

What is the input and output of the search? Is it image+text input? What data is returned?

**Limitations:**

Yes

**Strengths And Weaknesses:**

**Strengths**

This is a very important topic. Search is being used as a tool increasingly in LLMs as a way to expand their knowledge and reduce hallucinations across a variety of applications, from personal chat bots to coding agents. Comparatively less work has focused on multi-modal LLMs. The questions in this benchmark cover a variety of domains which is also essential.

**Weaknesses**

The first main weakness is the presentation. I am still quite confused about how the pipeline works. The explanation in the paper repeatedly relies on Figure 2, which is incomprehensible. The icons in it are completely meaningless. A figure like this needs to show actual examples going through the benchmark construction process, not icons that have no clear meaning. For example, “During annotation, we adopt a fact-first strategy for multi-domain questions (see Figure 2 (A2))”. Looking at Figure A2, I have no idea what this means. What is “divergence search” and is this term defined anywhere? What does the book with a hand on top of it mean? Or “As shown in Figure 2 (A1), NAVIGATE adopts a failure-driven data construction paradigm rather than a knowledge-centric one.” What is a “failure-driven data construction paradigm” and how does this contrast to a “knowledge-centric one” and how is this shown in A1? Moreover, the paper needs to have qualitative examples shown in the main text. What are some qualitative examples of samples that got rejected as the various stages of filtering? What are some easy, medium, and hard samples? I had to go digging into the appendix to find qualitative samples to get an understanding of the benchmark, where they are just listed without much organization. Table 2 also does not read well. There are two many numbers and it is hard to parse. Bolding should be used instead of underlining. Why does the second subtable not have any indication?

The second main weakness is that the difficulty is defined in an erroneous manner and moreover, their actual benchmark samples shown in the appendix contradict their difficulty characterization. The benchmark conflates the number of images in a sample with the requirement of using search to solve it. On the one hand, they say that “NAVIGATE further incorporates search necessity as an explicit factor in difficulty grading” but then when they define the difficulty tiers in the next paragraph, it is just based on single image, single domain -> multi-image multidomain, and multi-image multi-domain. The text references Figure 2 D1 but it does not help here. Where does the search come in at all? The current categorization does not have anything to do with whether search is required. I don’t agree with the premise that a single image example does not require search. Why not? In fact, looking at samples in the appendix *I find both single image examples that would require search to answer and multi-image search examples that can be solved without search*. So what is presented in the text is not truthful, as they say “For search-required questions, we additionally remove those solvable without external search”. I actually think the benchmark would be much better if it only contained single image examples split into those that require search and those that do not. Multi-image reasoning is an orthogonal skill to knowing whether search is needed to answer a question.

---

> ### Author Rebuttal · Authors · 2026-03-31
>
> **Q1: Figure 2 and the tables are difficult to interpret, and the main text lacks sufficient qualitative examples.**
> We thank the reviewer for this suggestion that the current Figure 2, with abstract icons, makes the construction pipeline harder to follow. We will replace icons with concrete examples, add representative positive and filtered-out cases, and clearly annotate the input and output of each stage. We will also revise the related tables to improve readability.
>
> **Q2: The definition of difficulty tiers is unclear, and the role of search necessity in difficulty grading is misleading.**
> Thanks for pointing out that the current writing may be confusing. In NAVIGATE, the main axis of difficulty is visual compositionality, i.e., from single-image to multi-image single-domain and then to multi-image multi-domain. By contrast, search necessity is not intended as an additional difficulty dimension, but as a constraint used to characterize the model’s tool-use decision-making ability within a fixed difficulty tier. We will clarify this explicitly in the final version.
>
> **Q3: Why is the single-image subset used as the search-free setting?**
> We agree that single-image questions may still require search, while multi-image questions may not. In NAVIGATE, we use the single-image subset as the search-free subset not because single-image questions are inherently search-free, but as a deliberate design choice. More specifically, NAVIGATE targets two abilities: when and how to search. For evaluating when to search, we use the single-image subset as a cleaner control setting with fewer confounding factors. It is therefore not treated as naturally no-search, but as an intentional subset for testing search restraint.
>
> **Q4: Why are some samples that appear to involve external-world knowledge still placed in the search-free?**
> We would like to clarify that “visually self-contained” is intended in an operational, rather than absolute, sense. For the single-image subset, our design goal is not to claim that such questions are unrelated to the external world in every possible sense, but under our evaluation protocol, introducing search typically does not provide positive utility and may even hurt performance because of noise.  During sample selection, we therefore consider not only whether a question appears to involve external information, but also whether introducing search brings benefit to solving it. If a sample appears to involve open-world knowledge but search does not help—and may even push the reasoning off track, we do not classify it as search-required. Instead, such samples are useful for evaluating whether the model performs unnecessary search. We will clarify this explicitly in the final version.
>
> **Q5: The meanings of “divergent search” and “failure-driven data construction” are not sufficiently clear.**
>  Divergent search refers to expanding along multiple candidate paths and information sources, rather than following a single retrieval path; in our setting, this means starting from an intermediate cue (e.g., “left-hander”) and branching into multiple possible domains to establish cross-domain associations(e.g.,"politics","arts"). Failure-driven data construction means that the dataset is built from model failure cases observed in realistic task settings, rather than from predefined knowledge points or manually designed common-sense questions. We will explain these terms and their roles in the construction pipeline more explicitly in the final version.
>
> **Q6: What MLLMs were used to collect the dataset samples?**
> Thanks for this question. We did not rely on a single model during data construction. Instead, we used failure logs from multiple mainstream MLLMs, including GPT, Gemini, Seed and so on, as candidate sources for sample collection to avoid overfitting to the failure patterns of a certain model. We will provide a more detailed description in the appendix of the final version.
>
> **Q7: What are the inputs and outputs of the search process?**
> We do not define a custom retrieval interface. Instead, we use each search-enabled model’s official built-in search API with default settings. The model takes the images, question, and fixed prompt as input, and its native search stack determines the query and returns web evidence. Thus, our evaluation targets end-to-end behavior in the model’s native search environment. We will clarify this more explicitly in the final version.

---

> > ### Author Rebuttal · Reviewer_PZ4x · 2026-04-02
> >
> > Some of my questions were clarified, but given a main weakness of the paper is presentation, it is hard to say that I think the paper should be accepted without seeing a final version. I feel that this draft is too unclear. I will raise my score to weak reject. I also think that my philosophical disagreements are not really resolved. For example, I do not find the answer to Q4 very satisfactory because they are essentially saying that my observations are correct but not true "operationally" which I take means on average. This is too subjective in my opinion and just looking at the qualitative examples, I feel that there is a risk that the benchmark is too noisy. There should not be so much ambiguity in the categorization.

---

> > > ### Author Response · Authors · 2026-04-07
> > >
> > > **Q1: The paper remains unclear in presentation, especially Figure 2.**
> > > We fully understand the reviewer’s concern and agree that presentation remains one of the main weaknesses of the current draft. In particular, the issue with Figure 2 is not merely that the icons are visually abstract, but that the figure, in its current form, does not provide a self-contained explanation of the data construction pipeline in the main text. As a result, the reader has to repeatedly rely on the appendix to reconstruct the benchmark design. We therefore understand why, without seeing a revised final version, the reviewer would be hesitant to place full confidence in the paper’s clarity.
> > > For this reason, our revisions in the final version will be structural rather than cosmetic. Specifically, we will:
> > > - replace the abstract icons in Figure 2 with real examples, rewriting it as a step-by-step, case-based pipeline;
> > > - explicitly show the input, transformation process, and output at each stage;
> > > - add representative retained samples, filtered-out samples, and examples from different difficulty tiers to the main text;
> > > - reformat Table 2 with clearer subgroup labels and stronger visual emphasis.
> > >
> > > Our goal is for the reader to understand the benchmark construction logic from the main paper itself, without having to rely on the appendix to infer the meaning of each icon or term.
> > >
> > > **Q2: Why is categorization not too subjective or noisy, especially when some examples seem counterintuitive from a human perspective?**
> > > We would like to further clarify that, by operational, we do not mean an “average-case” or intuition-based judgment. Rather, we mean a model-centered, protocol-driven categorization rule. More specifically, our concern is not whether a sample appears, from a human perspective, to involve external-world knowledge, but whether, from the model’s problem-solving process, search constitutes a necessary and consistently beneficial step.
> > >
> > > Accordingly, whether a sample is labeled as search-required is not determined by a single observation. Instead, it is determined by a joint protocol: a sample is classified as search-required only if it satisfies both (1) search sensitivity and (2) external-knowledge dependence. Otherwise, even if it appears on the surface to involve open-world knowledge, we do not label it as search-required. Such samples are instead useful for evaluating whether the model performs unnecessary search and whether search interferes with reasoning that could otherwise be completed successfully.
> > >
> > > In this sense, our categorization is not a vague, human-intuition-based interpretation, nor an empirical claim that holds only “on average,” but a reproducible classification under a fixed protocol. Therefore, the fact that some individual examples may appear counterintuitive from a human perspective does not imply that the benchmark itself is ambiguously categorized or excessively noisy. Rather, it reflects that human-intuitive categorization and model-behavior-based categorization do not always coincide, and NAVIGATE deliberately adopts the latter as its labeling principle. We also acknowledge the importance of human-intuitive consistency, and we will consider incorporating additional human–model agreement analysis in future dataset updates.

---

### Decision · Program_Chairs · 2026-04-30

**Decision:**

Accept (regular)

**Comment:**

This paper presents NAVIGATE, a benchmark centered on images as primary evidence for open-web search planning and multi-step reasoning. NAVIGATE contains 500 questions spanning 20 domains, ranging from single-image problems to complex multi-image and multi-domain tasks. Experimental results demonstrate that current SOTA methods struggle with NAVIGATE, with Gemini-3-Pro-Preview-Search reaching only 36.4% accuracy.

Reviewers praised the interesting setting of the dataset which has multiple images in web search and implicit cues derived from multiple images. They found experimental results interesting across domains and difficulty levels with SOTA models. Reviewers also commented positively on data curation and verification.

However, reviewers were concerned by the small scale of the dataset, as well as the lack of direct evaluation of search behavior (besides whether to search, how much to search, etc.), which leaves the experiments void of certain helpful metrics such as recall rate. Further, the automated evaluation method leaves the possibility of model bias.

All reviewers assessed the paper as weak accept, which I recommend.